# Fine-Tuning of DADA2 Parameters for Multiregional Metabarcoding Analysis of 16S rRNA Genes from Activated Sludge and Comparison of Taxonomy Classification Power and Taxonomy Databases

**DOI:** 10.3390/ijms25063508

**Published:** 2024-03-20

**Authors:** Wiktor Babis, Jan P. Jastrzebski, Slawomir Ciesielski

**Affiliations:** 1Department of Plant Physiology, Genetics and Biotechnology, University of Warmia and Mazury in Olsztyn, 10-719 Olsztyn, Poland; wiktor.babis@student.uwm.edu.pl (W.B.); bioinformatyka@gmail.com (J.P.J.); 2Department of Environmental Biotechnology, University of Warmia and Mazury in Olsztyn, 11-709 Olsztyn, Poland

**Keywords:** *16S rRNA* gene, activated sludge, biodiversity, DADA2, metabarcoding, OTUs

## Abstract

Taxonomic classification using metabarcoding is a commonly used method in microbiological studies of environmental samples and during monitoring of biotechnological processes. However, it is difficult to compare results from different laboratories, due to the variety of bioinformatics tools that have been developed and used for data analysis. This problem is compounded by different choices regarding which variable region of the *16S rRNA* gene and which database is used for taxonomic identification. Therefore, this study employed the DADA2 algorithm to optimize the preprocessing of raw data obtained from the sequencing of activated sludge samples, using simultaneous analysis of three frequently used regions of *16S rRNA* (V1–V3, V3–V4, V4–V5). Additionally, the study evaluated which variable region and which of the frequently used microbial databases for taxonomic classification (Greengenes2, Silva, RefSeq) more accurately classify OTUs into taxa. Adjusting the values of selected parameters of the DADA2 algorithm, we obtained the highest possible numbers of OTUs for each region. Regarding biodiversity within regions, the V3–V4 region had the highest Simpson and Shannon indexes, and the Chao1 index was similar to that of the V1–V3 region. Beta-biodiversity analysis revealed statistically significant differences between regions. When comparing databases for each of the regions studied, the highest numbers of taxonomic groups were obtained using the SILVA database. These results suggest that standardization of metabarcoding of short amplicons may be possible.

## 1. Introduction

Recently, due to the dynamic development of DNA sequencing techniques for analyzing environmental samples, it has become necessary to use a reliable method that allows for characterization of the organisms in those samples. One such method is metabarcoding, which allows the simultaneous identification of organisms and multiple taxa within the same sample using short fragments of DNA. To do this, researchers have searched for genetic barcodes that meet three conditions. First, they should have high genetic variability and differentiate organisms at the species level. Second, they should contain conserved flanking sites that allow the creation of universal PCR primers. Finally, they need to have the shortest possible sequences. For metabarcoding bacteria and archaea, the gene encoding the small subunit of ribosomes (*16S rRNA* gene) is usually chosen [1]. Nine variable regions (V1–V9) have been identified in the sequence of this gene, which allow the observation of interspecies differences [2]. There are also differences in variation between the regions themselves. In bacterial metabarcoding studies, the first five regions (V1 to V5) are commonly used. The lengths of the individual regions do not exceed the suggested values for barcodes, allowing them to be combined into fragments. In between these regions, fragments of low variability are present, which are also known as conservative regions. The region that proves to be most sensitive and accurate usually depends on the source of the sample.

Metabarcoding is usually used in environmental microbiology to classify bacteria at different taxonomic levels, detect the occurrence of taxonomic groups over time, seek differences between samples taken under different conditions, and determine biodiversity within or between samples. For example, the V3–V4 region was used to perform taxonomic classification and deeper analysis in studies that focused on tropical warm springs and different areas of the Rimac river [3,4]. In other examples, the V4–V5 region was used to study the microbial communities in rainwater and precipitation in a forest area and to compare five different ways to extract DNA from marine bacterial communities [5,6]. More and more often, metabarcoding is used to investigate technical biocenoses, such as activated sludge. Activated sludge is a flocculent culture of microorganisms and protozoa that develops in aeration tanks under controlled conditions and is used in biotechnological wastewater treatment processes [7]. For these studies, the combined V3–V4 region is most often used. In activated sludge studies, it is important to determine taxonomic groups, biodiversity, and taxon variability over time to monitor process performance [8].

Due to the fact that the results are obtained in the form of short readings, which are not always of appropriate quality, preprocessing is necessary. Preprocessing consists of such steps as adapter removal, read quality control, sequence filtering and trimming, merging, and chimera removal [9]. The choice of additional analyses depends on the specifics of the work: for the purpose of diversity analysis, applying alpha-diversity indices is usually enough. In the case where many samples are compared, beta- and gamma-diversity tests are used. Currently, for bioinformatic analysis, algorithms such as DADA2 [9] and Deblur [10] are applied. These algorithms are already part of open-source platforms, such as Galaxy or Qiime 2, that are used for taxonomic classification of microorganisms [11,12]. Several databases offering extensive collections of taxonomic groups may be encountered inside these platforms. A frequently used database is SILVA, which offers a high number of taxonomic groups and 99% identity [13]. We can also find databases such as Greengenes2 [14] or RefSeq [15].

However, the various approaches to metabarcoding often make it difficult to compare the results of different studies because, due to advances in NGS capabilities, the diversity of processes performed in laboratories and during bioinformatics analysis has expanded. When utilizing the same region of the *16S rRNA* gene, bacteria from different environments do not classify equally. In response to this problem, measures are being taken to standardize and optimize the entire process. One attempt at standardization is to examine the differences between taxonomic classifications using different regions in a particular environment to try to select the most accurate region for that environment [16]. Such standardization and the possibility of comparing results will certainly enable the development of metabarcoding processes in many fields.

Therefore, in this study using activated sludge samples, we wanted to attempt to standardize the process by optimizing preprocessing for metabarcoding using the three combined regions (V1–V3, V3–V4 and V4–V5) of the bacterial *16S rRNA* gene. The optimalization process included selecting parameters for the DADA2 plugin for Qiime2, with further complex control of the sensitivity and effectivity of the acquired operational taxonomic units (OTUs) from each tested combined region. Additionally, we performed a comparative analysis of three databases usually used for classification of bacterial taxonomies. 

## 2. Results

This study aimed to optimize the bioinformatic analysis of the microbial composition of activated sludge samples. As a first step towards achieving this goal, the preprocessing of raw sequences was optimized in the Qiime2 environment. Next, the effects of primers targeting different variable regions of the *16S rRNA* gene on indices of biodiversity was examined. Finally, the influence of using different databases on the classification of bacterial taxonomies was investigated.

The mean number of reads for the V1–V3 combined regions was 256,440 ± 16,293. The value for the V3–V4 combined regions was 214,490 ± 20,239, and for the V4–V5 combined regions, it was 160,356 ± 6954. 

The initial parameters for the DADA2 algorithm were set based on the QS results. This algorithm has three major steps: sequence filtration, sequence merging, and chimera removal. The results of each stage of the optimization process were averaged across all samples within each of the tested combined regions of the *16s rRNA* gene. The result of the initial filtration process was 52.84 ± 5.03% of the input reads from the V1–V3 combined region, but the percentage dropped drastically after merging and chimera removal to 0.77 ± 0.12% and 0.71 ± 0.12%, respectively (Figure 1A). Regarding the V3–V4 combined region, the drop from filtration, to merging, to chimera removal was not so drastic (70.92 ± 2.46%, 54.45 ± 4.24%, 37.28 ± 3.37%, respectively). As for the V4–V5 region, the drop from the initial to the final step was the smallest of the three regions (59.79 ± 2.60%, 54.45 ± 4.24%, 37.28 ± 3.37%, respectively). Based on these results, the optimization process first focused on increasing the output from both the V1–V3 region after the merging step and the chimera deletion step for all three regions. Changing --*p-min-overlap* from the default 12 to 10 increased the percentage of merged reads from the V1–V3 region to 6.54 ± 0.64%. Similarly, changing the --*p-chimera-method* from the default consensus to pooled increased the number of passed non-chimeric reads in all the tested combined regions by at least 1.5-fold (V1–V3: 6.36 ± 0.62%, V3–V4: 49.36 ± 3.66%, V4–V5: 52.93 ± 4.36%). In the second stage of optimization, changing the --*p-trunc-left-r* value increased the merging step of V1–V3, but decreased that of V3–V4 and V4–V5 (12.5 ± 3.44%, 17.3 ± 1.7%, 31.08 ± 1.45%, respectively); therefore, this change was not adopted.

Next, further decreasing the --*p-min-overlap* value to 8 was tested. This change increased the mean percentage of merged reads from V1–V3 to 7.54 ± 0.88%, without affecting the number of reads from the other regions. However, this percentage was still deemed too low; thus, in the fourth stage, the *--p-pooling-method* was changed from independent to a more sensitive option for rare variants—pseudopooling. Additionally, the value of *--p-trunc-len-f* was changed from 280 to 260 and the value of *--p-min-fold-parent-over-abundance* was changed from 1 (default) to 8. Unfortunately, even though these changes increased the mean percentage of the merging step in V3–V4 and V4–V5 (51.87 ± 3.72% and 54.93 ± 4.47%), they decreased that of V1–V3 (3.15 ± 0.25%). Therefore, in the fifth stage, the change from the fourth stage was reversed, and the value of --*p-trunc-len-f* from the initial 280 to 298 was changed. This modification increased the chimera deletion step of V1–V3 (17.55 ± 1.23%), while only slightly decreasing that of V3–V4 and V4–V5 (46.30 ± 3.73% and 49.80 ± 4.34%). Compared to the initial stage with the default values, these percentages were 14, 2, and 0.5-fold higher, respectively (Figure 1B). Further parameter modifications did not increase the final output. In the control, the effects of *--p-min-overlap*, *--p-pooling-method*, *--p-chimera-method*, and *--p-min-fold-parent-over-abundance* were investigated without trimming and truncating. Under these conditions, only 0.89% ± 0.57 of V1–V3 reads, 10.71% ± 2.27 of V3–V4 reads, and 3.41% ± 0.85 of V4–V5 reads successfully passed the filtration step in preprocessing.

The preprocessed reads were used to obtain the number of OTUs and to proceed to a comparison of the sensitivity and effectiveness of the combined regions. The average OTU count was the highest for the V3–V4 region with 6702 ± 402, followed by V1–V3 with 2563 ± 197, and V4–V5 with 1527 ± 58 OTUs. As shown in the correlation heatmap (Figure 2), there were negative, statistically significant correlations between the combined regions. The samples from the V1–V3 and V4–V5 regions displayed small negative correlations that were not statistically significant. Within each tested combined region, the correlations were positive and statistically significant.

Analysis of the combined areas revealed the highest values of Simpson’s index for the V3–V4 area, while the V1–V3 and V4–V5 areas had similar values (Figure 3A). There was a significant difference between V3–V4 and V1–V3 (*p* < 0.001), and between V3–V4 and V4–V5 (*p* < 0.01). There was no statistically significant difference between the V1–V3 and V4–V5 combined regions. 

As for the Chao1 index, the V4–V5 combined region exhibited the lowest value (Figure 3B). This value was significantly different from those of the V1–V3 combined region (*p* < 0.01) and the V3–V4 region (*p* < 0.001). In the case of the Shannon index, the V3–V4 combined region exhibited the highest value (Figure 3C). There was a statistically significant difference between V1–V3 and V3–V4 (*p* < 0.001), as well as between V3–V4 and V4–V5 (*p* < 0.01). After assessing diversity within each of the combined regions, beta-diversity was calculated using Bray–Curtis distance and visualized on a PCoA plot (Figure 4). There are noticeable differences between each combined region of the *16S rRNA* gene, but differences do not exist between samples in each tested region. The differences between combined regions were further confirmed with a PERMANOVA test, revealing a significant Pseudo-F value of 34.42 with a *p*-value of 0.001 for each combination between tested combined regions of *16S rRNA* gene.

In the last stage of the study, we conducted a comparative classification based on the three most popular databases. In the SILVA database, there were 510,508 taxonomy groups identified for the small subunit with a 99% identity threshold. For the RefSeq database, there were 25,974 taxonomy groups for Bacteria and 1140 for Archaea. Taxonomy of Greengenes2 is based on the Genome Taxonomy Database (GTDB) and Living Tree Project. The largest number of taxonomic groups was obtained by using the Silva database, and the smallest by using the RefSeq database (Figure 5). Differences in the number of taxonomic groups between databases within each combined region were statistically significant. The most unique taxonomic groups were obtained with the Silva database. It is worth pointing out that the unique taxonomic groups obtained from the RefSeq database were more numerous than those obtained from the Greengenes2 database. Differences in the number of unique taxonomic groups between databases within each combined region were statistically significant.

## 3. Discussion

In *16S rRNA* metabarcoding, determining an appropriate set of common parameters for the DADA2 algorithm for multiple regions of *16S rRNA* will provide an opportunity to reduce differences between various approaches to taxonomic analysis. Despite the popularity of this algorithm, suggestions for a set of parameter values adapted to a specific environment have still not been created. In addition, the use of ASV or OTU-based approaches is also a non-standard aspect. Although the use of ASVs is growing, activated sludge studies are often based on OTUs [17,18,19].

Therefore, this work suggests a possible set of parameters for the DADA2 algorithm with which to further standardize the metabarcoding process of *16S rRNA* gene amplicons and simplify the comparison of results. Obtaining a fully standardized process requires further work with larger samples, with which the sensitivity to adjustments of the algorithm’s parameters will be increased.

The choice of which variable region of the 16 s rRNA gene (V1 to V5) varies from one field to the next. In the environmental and biotechnology fields, the suitability of these regions was evaluated using the Chao1, Simpson and Shannon (alpha-diversity) indices, as well as beta-diversity indices and other analyses. For example, in a study using environmental samples, Bukin et al. [20] compared differences in microbial classification using the V2–V3 and V3–V4 regions. Using the previously mentioned biodiversity indices, it was indicated that the V2–V3 region showed greater resolution at the genus and species level. However, in a study by Brandt and Albertsen [21], it was shown that when comparing the sensitivity and power of taxonomic classification of the V1–V3, V3–V4 and V4 regions, the V3–V4 or V4 region has the highest power. In contrast, human origin samples frequently use the first regions of *16S rRNA*. An example is the study of respiratory samples, in which four regions were selected to compare regions and test their taxonomic identification ability: V1–V2, V3–V4, V5–V7 and V7–V9. In addition, because of the nature of the work and the possible use of the results, receiver operating characteristic (ROC) curves were additionally used. The result of this work was that the V1–V2 region is the most sensitive and has the highest power for taxonomic classification. Therefore, The Human Microbiome Project used the V3–V5 region as the core region for taxonomic classification, while the V1–V3 region was used to enhance the power of the resulting classification [22].

In the context of this difference between fields and due to the lack of comparison studies with activated sludge, it was unclear which region should be used for evaluating samples from wastewater treatment plants. This study helps to fill this gap in knowledge by suggesting that for biotechnological samples such as activated sludge, the V3–V4 region shows the greatest power to carry out taxonomic studies. It is noteworthy that parallel work with the other regions would fill in the missing OTUs.

In the metabarcoding approach, it is necessary to use an appropriate database for obtaining the most complete representation of microbial communities. Usually, for taxonomic classification of microbes the SILVA and Greengenes databases are used [23,24]. In the case of the Greengenes database, this was recently replaced by the new Greengenes2 in mid-2023. In addition, the RefSeq database created by NCBI (National Center for Biotechnological Information) is being expanded, strengthened, and used more and more often.

Finding that SILVA and Greengenes2 are the leading databases for taxonomic classification of microorganisms, this study suggests that for activated sludge SILVA is also the most appropriate choice. The same database classified the most specific taxa to low taxonomic levels (genus, species), which is required in the control of biotechnological processes. In the case of the RefSeq database (NCBI), this can compete with the Greengenes2 database, and it needs further development to be suggested as a leading database in bacterial taxonomic analyses.

Although many studies have taken up the problem of standardizing work with bacterial amplicon sequencing, it still requires continued analysis and work. To be able to take the next steps, more details of their work need to be made available, such as the possibility to view raw data, scripts of commands with their parameters, and full taxonomic classification lists. The use of a single or a combination of 2–3 variable regions may not be sufficient to obtain a complete list of taxonomic groups. It may be required to perform two simultaneous analyses using different regions. Another solution may be to use full-length analysis of the *16S rRNA* gene using nanopore sequencing.

## 4. Materials and Methods

Seven samples of activated sludge were obtained from the municipal wastewater treatment plant in Poznań, Poland (52.4493_N, 16.9826_E) treating municipal wastewater. Activated sludge samples (4 L) were collected in about 2-month intervals from the aerobic chamber. After sampling, activated sludge samples were placed at 4 °C and immediately transported to the laboratory, where they were frozen at −20 °C. DNA extraction samples were thawed at room temperature. DNA was extracted with a FastDNA Spin kit for soil (MP Biomedicals, Irvine, CA, USA), 200 mg of semi-dry biomass obtained by short centrifugation was re-suspended in a bead solution. Bead beating was performed at maximum speed in a Uniequip device (Uniequip, Planegg, Germany) for 5 min [25]. The quality of the extracted DNA was evaluated with agarose gel electrophoresis (0.8%), and the DNA concentration was measured fluorometrically with a Quant-iT BR DNA Assay (Thermo Fisher Scientific, Waltham, MA, USA) (Appendix A). 

To determine the optimal genome region for taxonomic composition analysis, three variable regions of the *16S rRNA* gene were tested (Figure 6).

The V1–V3 region was analyzed by amplification using the 27F (5′-AGA GTT TGA TYM TGG CTC AG-3′) and 534R (5′-ATT ACC GCG GCT GCTGG-3′) primers [26]. The V3–V4 region was amplified with the 341F (5′-CCT ACG GGN GGC WGC AG-3′) and 785R (5′-GAC TAC HVG GGT ATC TAA TCC-3′) primers [27]. Finally, the V4–V5 region was amplified using the 515F (5′-GTG CCA CCM GCC GCG GTA A-3′) and 944R (5′-GAA TTA AAC CAC ATG CTC-3′) primers [28]. Illumina adapters were added to all primers.

After sequencing, raw reads were checked using FastQC v0.12.0 [29] and the MultiQC tool v1.18 [30]. Sequence counts and the percent adapter content were checked. The fact that the maximum percentage of adapters did not exceed 1.08% indicated that adapter removal was not necessary and that their presence would not affect the quality of the analyses. For further filtration, the reads were imported to the QIIME 2 environment v2023.09. First, the quality score (QS) was checked using QIIME2 View (Appendix A). After that, it was decided to trim the nucleotides on the ends of reads which had less than 15 QS. 

The Divisive Amplicon Denoising Algorithm 2 (DADA2) pipeline was selected for preprocessing with the *qiime2 dada2 denoise-paired* command. After an initial stage, the optimalization process continued with 5 stages of optimalization (Figure 7). Several additional optimization attempts were made, but none of them improved the result, so the output from stage 5 was used for further analysis. Due to the poor result for the V1–V3 region in the initial stage, it was decided not to publish the taxonomic classification in the study, but to place it along with the classification from the final stage in the Appendix A. Additionally, the commands for the parameters were checked without adjusting trimming and clipping (0 and 300 for both directions). 

After conducting each stage, the number of filtered, merged, and non-chimera reads was expressed as a percentage of the number of raw reads. These values were checked using the Qiime2 View tool (Appendix A). For each tested combined region, the mean value of the samples was calculated with the corresponding standard deviation (SD). Additionally, the initial and final outputs were visualized using the ggplot2 package, v3.4.4 for R, v4.3.1 [31,32].

To find potential differences between various combined regions of 16S rRNA, a table of operational taxonomic units (OTU) was created with Qiime2 tools. The BIOM file was created using QIIME2, exported and opened using the *read.biom()* function from the rbiom package, v1.0.3 in R [33]. Then, the number of OTUs and the alpha diversity indices (Chao1, Simpson index, Shannon index) were calculated with the *alpha.div()* function from the same package. To test the statistical significance of differences between index values, the Kruskal–Wallis test was performed using the *kraskal.test()* function from the stats package, v3.6.2, in R, followed by Dunn’s post-hoc test with Benjamini–Hochberg *p*-value adjustment using the *dunn.test()* function from the dunn.test package, v1.3.5 [34]. For all tests, the criterion for significance was set at *p* < 0.05. The indices were visualized separately with ggplot2. Additionally, Spearman correlation coefficients were calculated with *cor.test()* and visualized with *ggcorrplot()* from the ggcorrplot package, v0.1.4.1 [35]. Testing differences between *16S rRNA* combined regions was based on beta diversity analysis. Statistical examination was performed with the *qiime diversity beta-group-significance* command from Qiime2 using the Bray–Curtis distance and visualized as a principal coordinates analysis (PCoA) plot with Vega v5.26.1 [36,37]. Finally, permutational multivariate analysis of variance (PERMANOVA with Bray–Courtis distances, permutations = 999) was calculated and visualized using the same command as the above analysis of beta diversity [38].

In the next stage, three databases were compared: Silva v138.1, Greengenes2 v2022.10, and RefSeq BioProjects 33175 and 33317. The prebuilt Silva and Greengenes2 datasets were used in the Qiime2 pipeline for taxonomy classification. The RefSeq database was imported to the Qiime2 taxonomy pipeline with the RESCRIPt plugin [39]. After taxonomic classification, all taxonomic groups within these three data sets were compared and visualized in R. Similarly, unique taxonomic groups, i.e., taxonomic groups that were classified as belonging to a specific genus, were compared and visualized. To calculate the statistical significance of differences in the number of taxonomic groups obtained with each database, the Kruskal–Wallis test was used, followed by Dunn’s post-hoc test with a Benjamini–Hochberg correction.

## Figures and Tables

**Figure 1 ijms-25-03508-f001:**
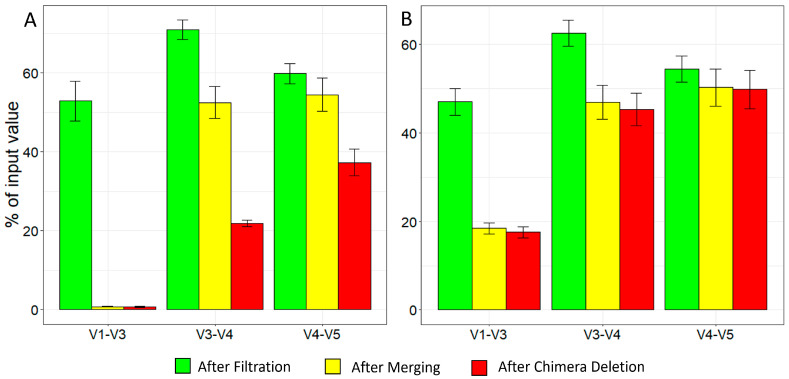
Average percentages of filtration, merging and non-chimeric read output from DADA2 pipeline. Each process from the DADA2 pipeline is represented by a different color. Each bar shows the mean percentage calculated from seven samples. (**A**) Mean percentages in each sample with initial parameters. (**B**) Mean percentages with parameters after the optimization process.

**Figure 2 ijms-25-03508-f002:**
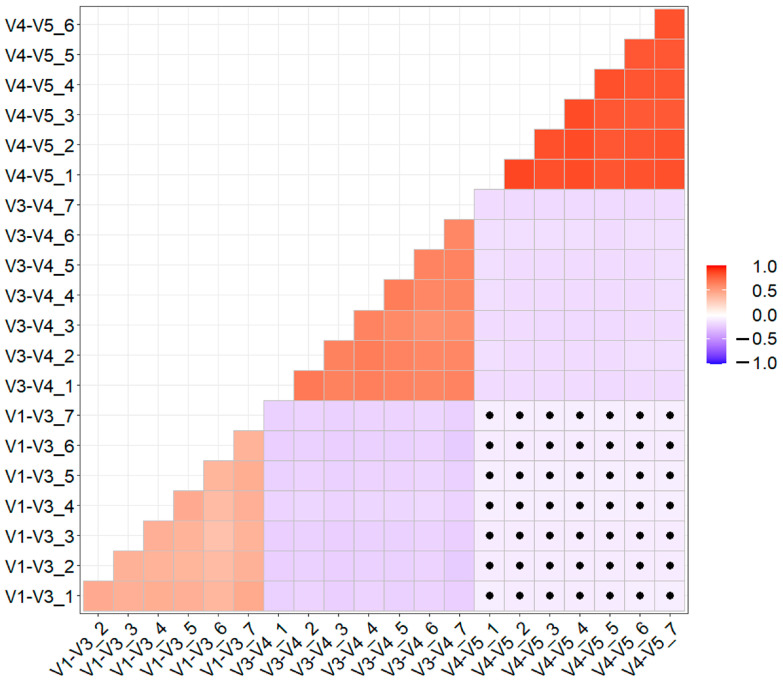
A heatmap depicting the Spearman correlation coefficients between each sample of tested fragments. The intensity of the color gradient reflects the direction and strength of the Spearman correlation. Black dots on the heatmap indicate non-significant correlation coefficients (*p* > 0.05).

**Figure 3 ijms-25-03508-f003:**
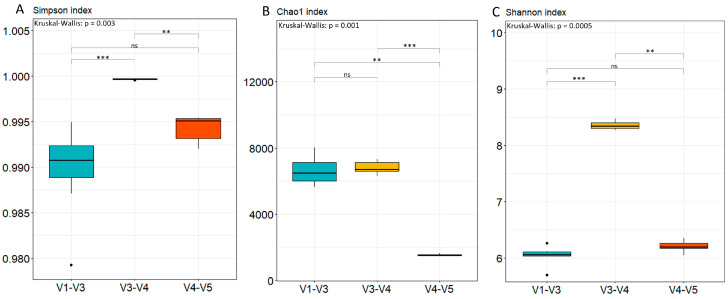
Visualization of alpha-diversity indices: Simpson index (**A**), Chao1 index (**B**), and Shannon index (**C**). Each of the tested *16S rRNA* combined regions is represented by a different color: V1–V3 (light blue), V3–V4 (yellow) and V4–V5 (red). Statistical significance was evaluated with the Kruskal–Wallis test followed by Dunn’s post-hoc test. A non-significant difference between combined regions is shown with “ns”, *p* < 0.01 is indicated by two stars (**), and *p* < 0.001 is shown by three stars (***).

**Figure 4 ijms-25-03508-f004:**
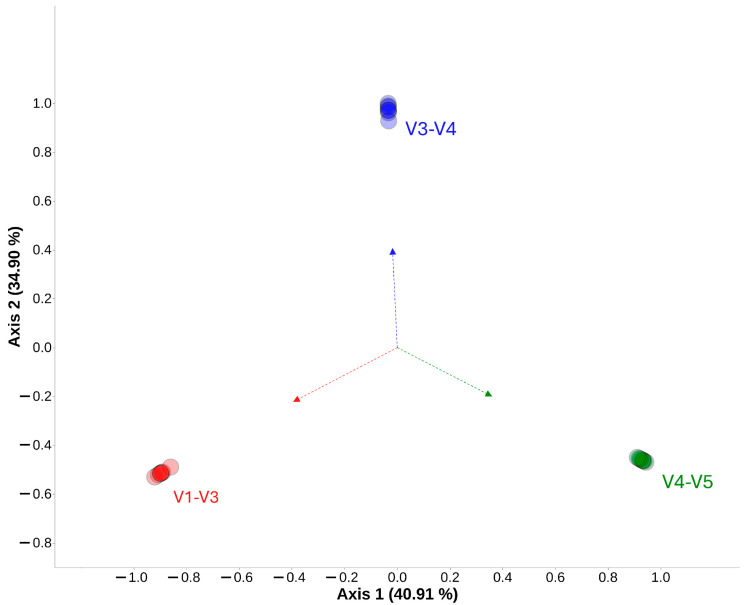
Beta-diversity visualized with principal coordinates analysis (PCoA) of seven samples from each tested combined region. Each color represents a different combined region: V1–V3 (red), V3–V4 (blue), and V4–V5 (green).

**Figure 5 ijms-25-03508-f005:**
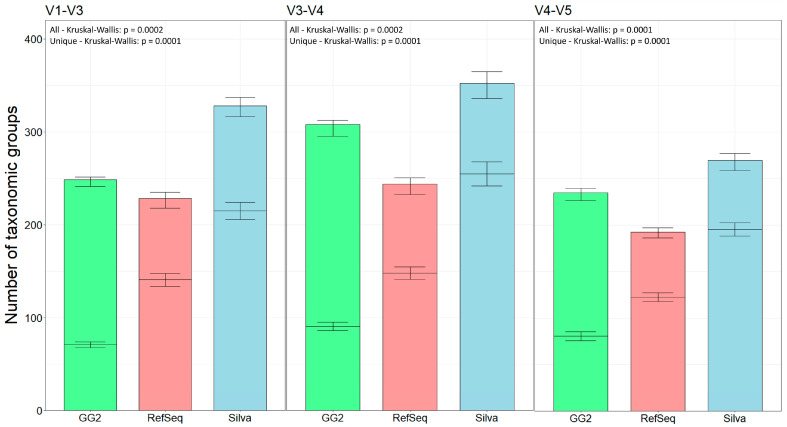
Mean quantities of all taxonomic groups obtained from three different databases within each region tested (*n* = 7). The Greengenes2 (GG2) database is shown in green, RefSeq is shown in red, and Silva is shown in blue. For all three regions, the values on the Y axis are the same. The differences of all taxonomic groups obtained in the taxonomic classification within the combined region are statistically significant. Inside each bar is a black dash indicating the number of unique taxonomic groups. Differences between the number of unique taxonomic groups within the combined region are statistically significant.

**Figure 6 ijms-25-03508-f006:**
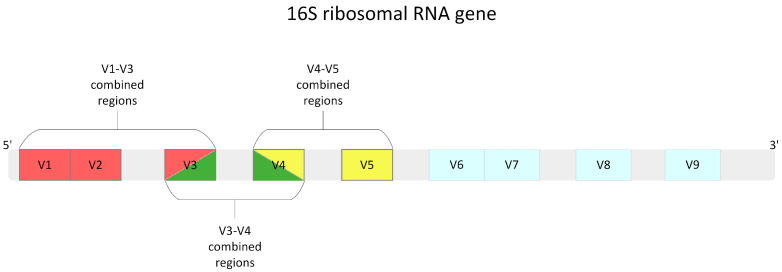
Diagram of the *16S rRNA* gene showing variable regions. Highlighted fragments with red, green, and yellow color were examined in the analysis. Light grey indicates a conservative region in *16S rRNA*.

**Figure 7 ijms-25-03508-f007:**
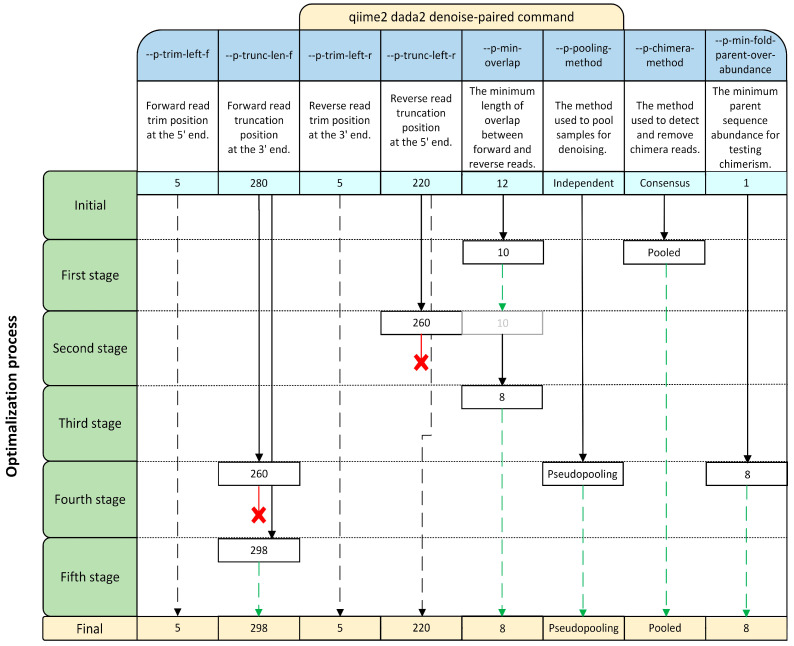
Stages of the optimization process with the qiime2 dada2 denoise-paired command in the Qiime2 pipeline. Black solid arrows indicated changes between a stage and the initial parameters. Black dashed arrows show parameters with final values that were unchanged from the initial ones. Red lines with an X indicate changes that decreased the percentage of reads that were obtained and were not retained in further stages. Green dashed arrows correspond to changes that increased the percentage of reads.

## Data Availability

Data will be made available on request.

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
