# Peer review of "Fine-Tuning of DADA2 Parameters for Multiregional Metabarcoding Analysis of 16S rRNA Genes from Activated Sludge and Comparison of Taxonomy Classification Power and Taxonomy Databases"

_ijms, 2024, doi:10.3390/ijms25063508_

Round 1

Reviewer 1 Report

Comments and Suggestions for Authors

1.      In Material and methods : Sample Collection details are missing which include geographical location details, Activated sludge sample details, Sample transferring details to laboratory, sample storing or preserving temperature details before metabarcoding.

2.      In Line 276: Fast DNA spin kit point has to come after bead solution point.

3.      Line 279: agarose gel electrophoresis -  Agarose concentration?

4.      Line 279: DNA concentration – Results?

5.      Line 54: How far have you validated this example - For example, the V3-V4 region was used to perform taxonomic classification and deeper analysis in studies that focused on tropical warm springs and different areas of the Rimac river. In other examples, the V4-V5 region was used  to study the microbial communities in rainwater and precipitation in a forest area.

6.      Line 74: Why you have compared with 3 different databases SILVA, Greengenes2, RefSeq?

7.      Line 300 to 314: Why detailed commands are given? Is there any specific reason? Or you can just give those commands in supplementary data.  

Author Response

Dear Reviewer,

Reviewer 2 Report

Comments and Suggestions for Authors

The presented manuscript "Fine-Tuning of DADA2 Parameters for Multiregional Metabarcoding Analysis of 16S rRNA Genes from Activated Sludge and Comparison of Taxonomy Classification Power and Taxonomy Databases" present study for parameter optimization for DADA2 software.

The manuscript is well written, however there is some suggestions the autors need to adress before it is considered for publication:

Major comments:

The authors should include the outcome results of the sample analysis and before and after the optimization as a taxonomical results and also discuss the actual relevant biological outcome as a taxonomy.

Minor comments:

Line 55 the reference is in wrong format

Author Response

Dear Reviewer,
